# Design and Experiment for N = 3 Wineglass Mode Metal Cylindrical Resonator Gyroscope Closed-Loop System

Xiaolin Guo, Rang Cui, Shaochen Yan, Qi Cai, Wenqiang Wei, Chong Shen and Huiliang Cao *

Key Laboratory of Instrumentation Science and Dynamic Measurement, Ministry of Education,
North University of China, Taiyuan 030051, China
* Correspondence: caohuiliang@nuc.edu.cn

**Abstract:** This paper studies a kind of gyro structure of N = 3 Wineglass Mode Metal Cylindrical Resonator Gyroscope (WMMCRG). Compared with traditional Cylindrical Vibrating Gyroscope (CVG), the designed structure has higher scale factor and lower frequency split. This paper provides a more specific processing method and the parameters of resonator materials. A closed-loop controlling system with low error and low noise is designed for WMMCRG. The system is composed of three independent closed-loop systems: drive closed-loop, sensing closed-loop, and quadrature error correction closed-loop. Through the test of the high-precision turntable, under the premise of the same material and processing technology, the bias instability, bias stability, zero bias, Angular Random Walk (ARW), and frequency split of WMMCRG is $1.974°/h$, $10.869°/h$, $10.3323°/s$, $16 (°)/\sqrt{h}$, $0.02$ Hz, respectively.

**Keywords:** cylindrical resonator gyroscope; control method; wineglass mode

## 1. Introduction

According to their working principles, gyroscopes can be divided into three categories: mechanical rotor gyroscopes, optical gyroscopes, and vibrating gyroscopes. The Coriolis vibrating gyroscope is one of the fastest growing and most commonly used gyroscopes in inertial navigation systems, in which the hemispherical resonator gyroscope (HRG) is more mature and has been well applied. Quality factors and frequency matching play an important role in the performance improvement of resonators, which determines the noise level and sensitivity of Coriolis vibrating gyroscope. The Q factor and bias stability of the HRG represented by the Northrop Grumman Company of the United States has reached 25 million and 0.0001/h, and can work continuously for 16 million hours without failure. Despite its remarkable advantages, HRGs have a number of problems, including extremely high cost and a complex manufacturing process.

In [1], an optimal quadrature error correction scheme is proposed for the dual-mass MEMS gyroscope. In light of utilizing the Coupling Stiffness Correction (CSC) technique, the Angle Random Walk (ARW) can be increased from $0.66°/\sqrt{h}$ to $0.21°/\sqrt{h}$. In [2], a strategy, named the sensing mode force rebalancing combs stimulation method (FRCSM), is presented to simulate the Coriolis force. This method expands the gyro bandwidth from 13 Hz to 102 Hz. In [3], three methods, including radial basis function neural network (RBF NN), RBF NN based on genetic algorithm (GA), and RBF NN based on GA with Kalman filter (KF), are proposed to increase three-axis MEMS vibration gyro performance efficiency. The bias instability of Gyros X, Y, and Z improve from $139°/h$, $154°/h$, and $178°/h$ to $2.9°/h$, $3.9°/h$, and $1.6°/h$, respectively.

Note that the abovementioned studies [1–10] are incapable of avoiding frequency split, which always occurring in the manufacturing process, obviously reducing the performance of gyroscope. In addition, it should be pointed that the existing research [1–10] does not pay much attention to increasing the mechanical sensitivity by regulating the structure, which limits the development of gyroscope. Therefore, based on the abovesaid observation, this

paper proposes a new structure of gyroscope, which can obviously reduce the frequency split and improve mechanical sensitivity.

Under the same processing technology, the frequency split of the N = 3 Wineglass Mode Metal Cylindrical Resonator Gyroscope (WMMCRG) is about 96% smaller than that of Cylindrical Vibrating Gyroscope (CVG), which are 0.02 Hz and 0.6 Hz, respectively [11–22].

The contact driving method is adopted, and specially designed tooling is used to locate the piezoelectric chip to ensure that all the piezoelectric pieces are evenly distributed on the bottom of the gyroscope [23]. The glue quantity is controlled by the glue dispensing machine; the minimum resolution is 0.01 mL and the minimum glue thickness is 560 um on the piezoelectric chip with an area of 18 mm$^2$.

This paper provides the manufacturing method of a low-cost Coriolis vibrating gyroscope with an extremely low frequency split. The tests of the WMMCRG show that the bias instability of the WMMCRG is 1.974 (°)/h, and the angle random walk is 16 (°)/√h. However, this technology is temporarily limited to the smaller CVG, and it is not known whether it is suitable for other kinds of gyroscopes [24–26].

The aim of this paper is to study the parameters of the WMMCRG after closed-loop control, and requires the controller to have simple structure and good adaptability. After adopting the new structure, the performance of the gyroscope will be improved obviously. Section 2 introduces the structure of the WMMCRG, including modal analysis, dynamic equations, and a structural simulation. Section 3 introduces the processing technology of the WMMCRG and the installation of the electrode in detail, and provides the specific heat treatment and electrode installation scheme. In Section 4, the control method of the WMMCRG and the design idea of the controller are introduced. Then, in Section 5, the open-loop and closed-loop of the WMMCRG are tested, and the final static test results are shown. Finally, Section 6 discusses the experimental results and comments on the conclusions.

## 2. N = Three Wineglass Mode Metal Cylindrical Resonator Gyroscope (WMMCRG) Sensitivity Structure

The WMMCRG structure diagram investigated in this paper is shown in Figure 1. It can be seen from Figure 1 that the structure is centrally symmetrical; 12 beams are uniformly distributed on the bottom of the gyro; the beam is the place where the piezoelectric electrode are pasted; and the droplet-shaped holes between the beams play the role of stress concentration and positioning. Six discrete piezoelectric electrodes are distributed in two degenerate modes, of which two electrodes are used as drive and four electrodes are used for detection [2,13,23]. There are a total of four drive electrodes and eight detection electrodes on the gyro, Figure 2.

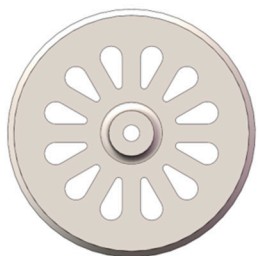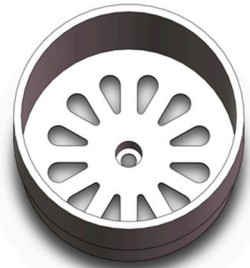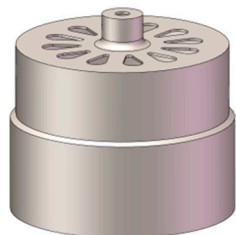

**Figure 1.** The structure of WMMCRG.

DX (+) and DX (−) are the electrodes of the driving mode, and DX (+), DX (−) applies sinusoidal signals with a phase difference of 180°. SX (+) and SX (−) are used as driving feedback electrodes, and the signal is amplified and filtered into the preprocessing circuit. When the gyro has angle signal input, the sensing electrode SY (+), SY (−) picks up the Coriolis force signal, and the gyro rotational speed signal is obtained by processing. DY (+) and DY (−) electrodes are quadrature error correction electrodes and sensing closed-loop electrodes [22].

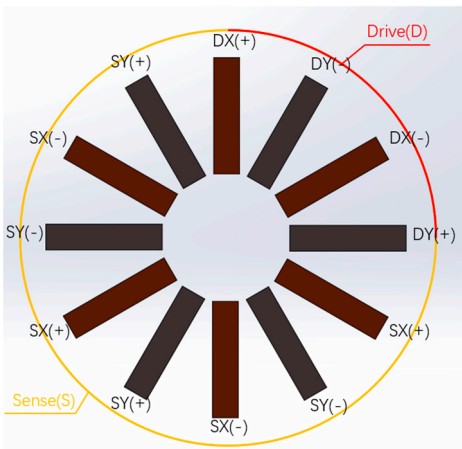

**Figure 2.** The electrode distribution of the gyro, of which four are drive eletrodes and eight are sense electrodes.

The first eight order modes of the structure are simulated and shown in Figure 3, and the working modes are emphasized. The resonant structure of the metal shell designed in this paper is classified according to half *n* (the shell vibration displacement is zero) of the number of nodal lines. The modal characteristics of the metal shell resonant structure designed in this paper are shown in Figure 3.

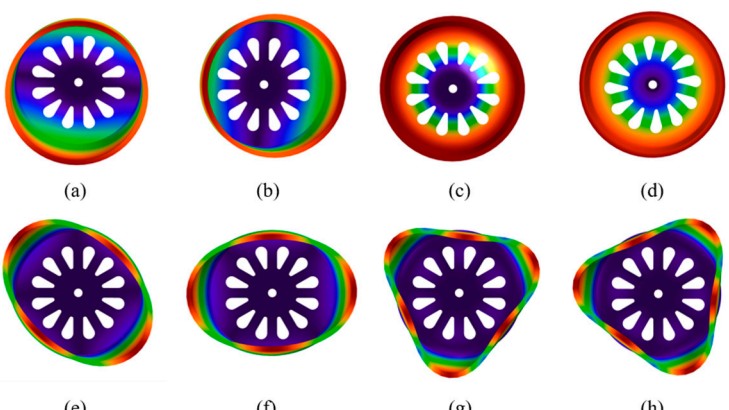

**Figure 3.** WMMCRG structure lower eight vibration order: (**a**,**b**) swing mode (204.36 Hz, 204.49 Hz); (**c**) vertical vibration modes (570.97 Hz); (**d**) Z axis in phase twist mode (2524.78 Hz); (**e**,**f**) wineglass modes (n = 2) (3057.6 Hz, 3057.7 Hz); (**g**,**h**) wineglass modes (n = 3) (8065.9 Hz, 8065.9 Hz).

The first (Figure 3a) and second modes (Figure 3b) of the gyro are swing modes (n = 1). The shell structure vibrates around the center of the anchor point, and the swing mode is one of the degenerate modes, and the difference between the two modes is 90°. The third (Figure 3c) and fourth modes (Figure 3d) are vertical vibration modes (n = 0), and the shell moves as a whole along the central axis and twist. The fifth (Figure 3e) and sixth modes (Figure 3f) are wineglass modes (n = 2), which are "round-oval" four-wave belly vibration; the two modes are 45° to each other. The seventh (Figure 3g) and eighth modes (Figure 3h) are working modes (n = 3). The shell mode is completely symmetrical around the symmetry axis as a "round-equilateral triangle" six-wave belly vibration state, and the two modes are 30° to each other, which are the driving mode and detection mode, respectively.

The fifth and sixth modes and the seventh and eighth modes are in complete dynamic equilibrium with respect to the symmetric axis, and the frequency difference in the seventh and eighth modes is simulated as 0 Hz, so the six-wave mode is selected as the working mode of the WMMCRG. At the same time, the two working modes are typical inverse dif-

ferential motions, which can effectively restrain the influence of common mode acceleration and have obvious advantages in environmental adaptability.

The vibration mode of the resonator has the influences of of frequency split, non-uniform damping, mass imbalance, and other factors. In this paper, the mathematical model of the hemispherical resonant gyro derived by Lynch is used to analyze the WMMCRG error propagation equation [24–26]. The WMMCRG two-dimensional vibration equation with error term derived by Lynch can be expressed as [27]:

$$
\begin{cases}
\ddot{a} - 6k\Omega\dot{b} + (\frac{2}{\tau} + \Delta(\frac{1}{\tau})\cos 6\theta_\tau)\dot{a} + \Delta(\frac{1}{\tau})\sin 6\theta_\tau\dot{b} \\
\quad + (\omega^2 - \omega\Delta\omega\cos 6\theta_\omega)a - \omega\Delta\omega\sin 6\theta_\omega b = f_a \\
\ddot{b} + 6k\Omega\dot{a} + (\frac{2}{\tau} - \Delta(\frac{1}{\tau})\cos 6\theta_\tau)\dot{b} + \Delta(\frac{1}{\tau})\sin 6\theta_\tau\dot{a} \\
\quad + (\omega^2 + \omega\Delta\omega\cos 6\theta_\omega)b - \omega\Delta\omega\sin 6\theta_\omega a = f_b
\end{cases}
\tag{1}
$$

In the Formula (1), $a$ is the direction of the gyro drive axis and $b$ is the direction of the detection axis, where:

$$
\omega^2 = \frac{\omega_a^2 + \omega_b^2}{2}, \frac{1}{\tau} = \frac{1}{2}(\frac{1}{\tau_a} + \frac{1}{\tau_b}), \omega\Delta\omega = \frac{\omega_a^2 - \omega_b^2}{2}, \Delta(\frac{1}{\tau}) = \frac{1}{\tau_a} - \frac{1}{\tau_b}
$$

$\omega_a$ and $\omega_b$ are the resonant frequencies of two vibration modes (shown in Figure 4.). Due to the existence of frequency splitting, these two frequencies are not equal. It is assumed that $\omega_b < \omega_a$, $\theta_\omega$ is the azimuth between the rigid "frequency normal axis" $\omega_b$ and the X axis. Similarly, $\tau_a$ and $\tau_b$ are the time attenuation constants of two "damping normal axes". It is assumed that $\tau_a < \tau_b$, $\theta_\tau$ is the azimuth between the "damping normal axis" of $\tau_a$ and the X axis. The items in equation are described as follows:

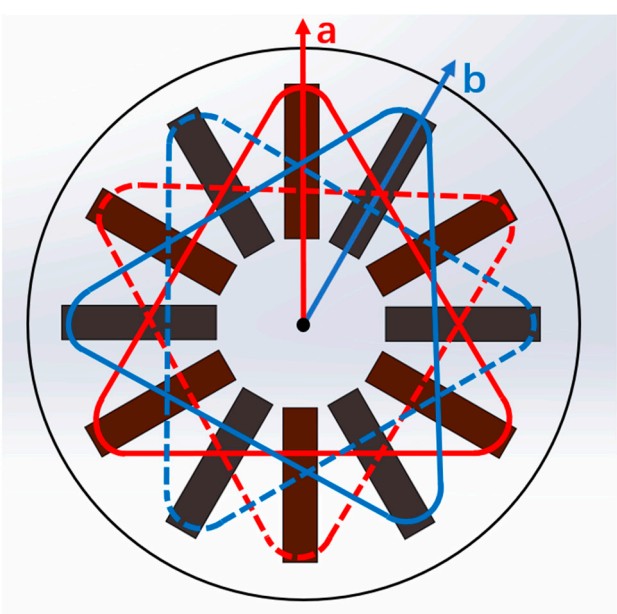

**Figure 4.** Axis of drive (**a**) and sensing (**b**).

1. The $-6 k\Omega\, \dot{b}$, $6 k\Omega\, \dot{a}$ term is the gyro effect term caused by the Coriolis force under the condition of gyro base rotation, which belongs to the useful signal.

2. $\frac{2}{\tau}\dot{a}$, $\frac{2}{\tau}\dot{b}$ is the attenuation term caused by the co-directional damping of the resonator.

3. $\Delta(\frac{1}{\tau})\cos 6\theta_\tau\dot{a}$, $\Delta(\frac{1}{\tau})\sin 6\theta_\tau\dot{b}$, $-\Delta(\frac{1}{\tau})\cos 6\theta_\tau\dot{b}$, $\Delta(\frac{1}{\tau})\sin 6\theta_\tau\dot{a}$ is an error coupling term caused by non-codirectional damping.

4. $-\omega\Delta\omega\cos 6\theta_\omega a$, $-\omega\Delta\omega\sin 6\theta_\omega b$, $\omega\Delta\omega\cos 6\theta_\omega b$, $-\omega\Delta\omega\sin 6\theta_\omega a$ is the error coupling term caused by frequency splitting.

5. $f_a$, $f_b$ is the force applied to the resonator by the control electrode.

## 3. WMMCRG Resonator Processing and Electrode Assembly

The metal structure of the WMMCRG resonator is a typical thin-walled part of rotation, and its material characteristics, machining accuracy, and machining efficiency are the key to determining the high performance and low cost of the gyroscope. According to the requirement of full symmetry of the resonator structure of the cup-shaped undulatory gyroscope with high sensitivity, the high shape, position accuracy, and surface quality should be guaranteed in the precision machining process of the resonator metal structure. In addition, the high Q factor and good temperature stability of the WMMCRG also put forward high requirements for material selection and heat treatment of the resonator metal structure [28–30].

The resonator material is alloy 902 of Ni-Span-C Company of the United States, and its composition and performance parameters are as follows: Ni (41–43.5%), Cr (4.9–5.7%), Ti (2.2–2.7%), Al (0.3–0.8%) Mn < 0.8% science C < 0.06%, and the rest is Fe. Working temperature range: $-45\sim70$ °C, linear thermal expansion coefficient: $7.6 \times 10^{-6}/$°C, mechanical quality factor: $\geq$20,000, frequency temperature coefficient: $-5\sim5 \times 10^{-6}/$°C.

What has a greater impact on the properties and uses of alloy materials is the heat treatment process of materials, because the characteristic of heat treatment, or deformation heat treatment of constant elastic alloy, is that they not only strengthen the alloy, but also shape the elastic elements. What is more important is to use the heat treatment process to adjust the distribution of elements between matrix metals and metal compounds, and control the structure and structure of the alloy. Finally, the requirements of the frequency temperature coefficient and the mechanical quality factor of resonator are realized. In this paper, the heat treatment process of the resonator metal material mainly includes the following links [30]:

(1) Solid solution treatment. If the cold deformation and aging treatment are to be carried out, Ni-Span-C alloy902 needs to be treated by solid solution first to obtain a supersaturated single solid solution at room temperature. The grain size of the alloy increases with the increase in solution treatment temperature below 1000 °C, but when the solution temperature is above 1000 °C, the grain grows rapidly, resulting in the decrease in alloy plasticity and in the difficulty of mechanical processing.

(2) Cold deformation treatment. In order to increase the lattice distortion and dislocation density of Ni-Span-C alloy902 and improve its strength, it is necessary to carry out cold deformation treatment on Ni-Span-C alloy902. When the deformation degree reaches 50% to 60%, the strength change of the alloy is not obvious. For frequency elements used in dynamic applications such as a resonator, cold deformation is also one of the effective methods to improve the mechanical quality factor of alloys.

(3) Aging treatment. In general, the optimum aging temperature corresponding to the optimum frequency temperature coefficient, the maximum strengthening effect, and the highest quality factor of Ni-Span-C alloy902 alloy are different. Therefore, according to the performance requirements of the harmonic oscillator, some strength performance indexes should be sacrificed in the heat treatment process to ensure the low frequency temperature coefficient and high mechanical quality factor as far as possible. The preliminary machined resonator is shown in Figure 5.

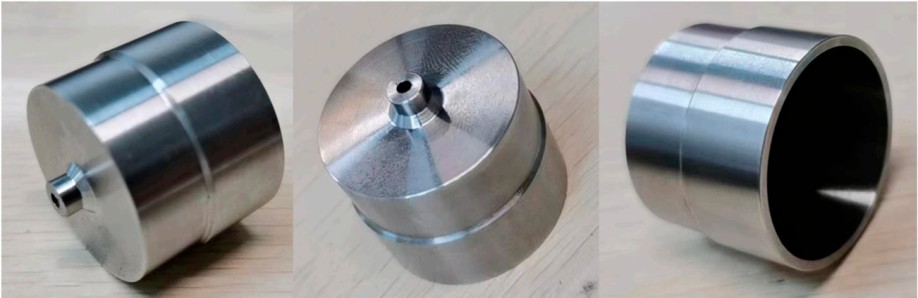

**Figure 5.** Photo of resonator after rough machining.

The resonator is machined by DJC-100E ultra-precision single-point diamond lathe; the maximum diameter can reach 100 mm, the surface roughness of the workpiece (Ra) < 2 nm, the surface shape precision PV value <0.1 um, the three-axis resolution is 1 nm, and the straightness is 0.1 um/100 mm. Finally, the surface roughness Ra of the machined resonator is 0.085 um, and the roundness < 0.1 um and axiality < 0.1 um measured by Taylor Hopson profiler. The finished resonator is shown in Figure 6.

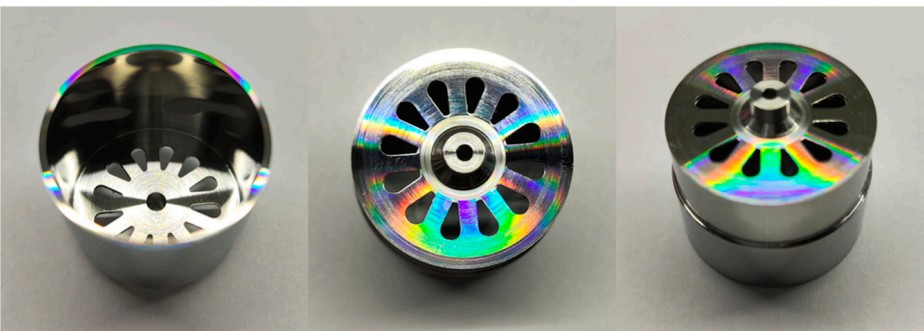

**Figure 6.** Physical photos of WMMCRG resonator from different angles.

The pasting process of the piezoelectric electrode is shown in Figure 7. The twelve slots of the positioning base are used to ensure the positioning accuracy of the piezoelectric electrode.

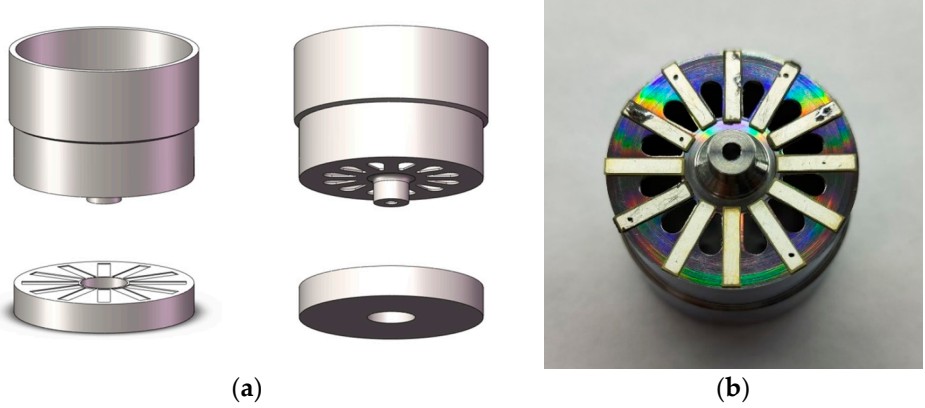

(**a**)                                          (**b**)

**Figure 7.** Piezoelectric electrode assembly tooling (**a**) and Photo of electrode assembly completed (**b**).

The piezoelectric electrode is fixed by springs and bolts; the force between the resonator and the positioning base is controlled by adjusting the number of bolts and the hardness of the spring; and the thickness of the glue layer is controlled by a gluing machine. The piezoelectric electrode mainly depends on the bonding layer to stick to the bottom of the resonator. The vibration of the piezoelectric electrode under alternating voltage is transmitted to the resonator through the bonding layer. Therefore, the quality of the bonding layer is very important for the performance of the gyroscope.

## 4. Circuit Design and Simulation

As a kind of resonator based on the piezoelectric effect, the WMMCRG resonator has both input voltage and output voltage on its piezoelectric electrode. The resonator itself is an elastic body with an infinite number of modes. The dynamic characteristics of each mode are similar to those of the RLC circuit, which are the dynamic characteristics of the second-order system. Therefore, the response characteristics of the WMMCRG resonator to the input voltage frequency can be described by the equivalent circuit model, as shown in Figure 8.

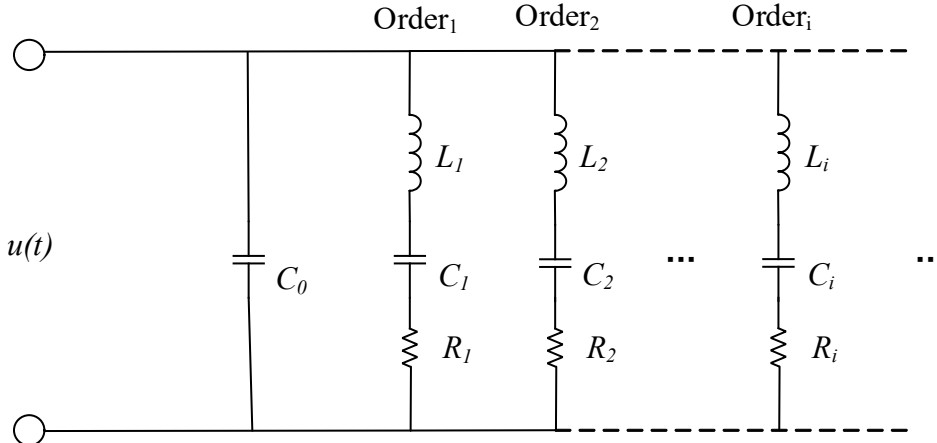

**Figure 8.** Equivalent Circuit Model of WMMCRG resonator.

Each branch in the equivalent circuit model describes a vibration mode of the resonator, in which $C_i$ and $L_i$ are the equivalent capacitance and equivalent inductance of the resonator, also known as dynamic capacitance and dynamic inductance, which are related to the mechanical properties of the resonator, corresponding to the order$_i$ modal mass and modal stiffness, respectively. $R_i$ is the equivalent resistance of the resonator, which is related to the mechanical loss of the material, and corresponds to the order$_i$ modal damping. In addition, the parallel capacitance $C_0$ represents the electrostatic capacitance between the piezoelectric electrodes. Although the dynamic response of the WMMCRG resonator is the superposition of an infinite number of mode responses, for the working mode of the WMMCRG with high quality factor, the resonator is near its working mode frequency, and the higher accuracy of the dynamic response can be obtained only by using the working mode of the resonator to analyze the steady-state response of the driving force. Therefore, the vibration of the working mode of the voltage excited resonator can be simplified to an RLC branch to be equivalent.

The gyro control loop is shown in Figure 9. In the drive closed-loop mode, the AGC control technology is employed, and the driving displacement is detected by the piezoelectric electrode SX and measured by a differential amplifier. Then, the phase of the signal is delayed, to satisfy the phase requirement of the AC drive signal $V_{dac}Sin$. Then the amplitude of the signal is obtained by full-wave rectification and low-pass processing of the $V_{dac}Sin$. The $V_{dac}$ is then compared with the reference signal $V_{ref}$. Next, the closed-loop drive PI controller generates a control signal and multiplies it with the $V_{dac}Sin$ to stimulate the drive mode. Based on the above closed-loop driving mode, the gyro driving mode can be stimulated at the resonant frequency with a steady amplitude, which can be set by $V_{ref}$.

The sense mode loop employs the same interface as the drive mode circuit. First, the movement signal of resonator is detected with differential detection amplifiers as $V_{stotal}$. Then, $V_{stotal}$ is demodulated by the signal $V_{dac}Sin$, and a demodulated signal $V_{dem}$ is generated. Then, $V_{dem}$ passes through the low-pass filter and forms the sense mode open-loop signal $V_{Oopen}$. For the sense feedback loop, $V_{Oopen}$ is sent to PIPAC to generate the controlling signal. Finally, the signal is modulated with $V_{dac}Sin$ to form the final feedback signal.

The quadrature error correction loop also uses the closed-loop system with the same mode. first, the quadrature error demodulation is used to pick up the SY signal into the demodulator, and the drive mode displacement of the drive closed loop is demodulated with the in-phase signal. Then the quadrature error compensation is carried out, and the demodulated signal is processed by the low-pass filter and compared with reference signal $V_{Qref}$ through the comparator. Finally, the quadrature error correction controlling signal is generated, and the compensation result is sent to PI controller. The resulting control signal is sent to the quadrature correction electrode to correct the quadrature error signal.

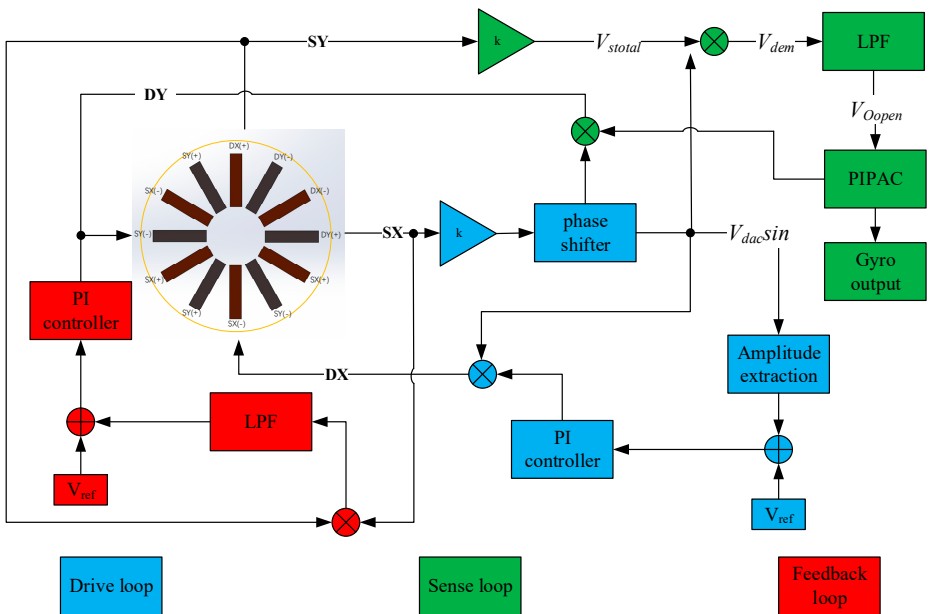

**Figure 9.** WMMCRG monitoring system schematic.

## 5. Experiment

### 5.1. Experiment Platform and Equipment

Prototypes of the WMMCRG were tested with atmosphere packaging to evaluate their resonance and performance characteristics. A sample of the WMMCRG (shown in Figure 8) was used to construct a simple working mode frequency response test system in a laboratory environment. The function signal generator generated an alternating voltage signal to be applied to the WMMCRG drive electrodes, and then the output voltage signal of the drive feedback electrodes was amplified by an amplifier circuit and then measured by a multimeter (Keysight 34401A, Santa Rosa, CA, USA).

The frequency test signal is generated by the function signal generator, and the frequency of the applied signal and the voltage amplitude detected by the multimeter are recorded. The maximum voltage value corresponded to signal frequencies of 8065.52 Hz with the amplitude of 2Vpp (drive mode), and 8065.5 Hz with the amplitude is 2Vpp (sense mode), which are basically consistent with the theoretical model data.

### 5.2. Scale Factor Test

A set of angular rates are inputted to the tested gyroscope by the rate turntable. The output values of the gyroscope are recorded under each input, and the scale factor of the gyroscope is obtained by least square fitting.

The test method is as follows:

a. Power on the gyroscope, set the sampling interval and sampling times $N$, and start the test after the preparation time.

b. The input angular rate of the turntable changes in the order of absolute value from small to large. The output $U_{ip}$ of the gyroscope is recorded at each angular rate, and the average $U_i$ of the output of the gyroscope at the input angular rate is obtained according to the Formula (2).

c. Enter no less than 11 points of angular velocity within the measurement range, which must include the maximum and minimum values of the measurement range.

The average $U_i$ of the gyro output is calculated according to the Formula (2):

$$U_i = \frac{1}{N}\sum_{p=1}^{N} U_{ip} \tag{2}$$

In the Formula (2), $U_i$ is the average output of the gyroscope with the $i$ angular rate input, $U_{ip}$ is the $p$ output value of the gyroscope with the $i$ angular rate input, and $N$ is the sampling number.

For the linear model of input and output of gyroscope, as Formula (3)

$$U = KX + b \tag{3}$$

In the Formula (3), $U$ is the output value of gyroscope, $K$ is the scale factor, $X$ is the input of gyroscope, and b is the zero position of gyroscope fitting.

The straight line is fitted by the least square method, and $K$ and $b$ are calculated according to the Formulas (4) and (5).

$$K = \frac{\sum\limits_{i=1}^{n} X_i U_i - \frac{1}{n} \sum\limits_{i=1}^{n} X_i \sum\limits_{i=1}^{n} U_i}{\sum\limits_{i=1}^{n} X_i^2 - \frac{1}{n} \left(\sum\limits_{i=1}^{n} X_i\right)^2} \tag{4}$$

$$b = \frac{1}{n} \sum\limits_{i=1}^{n} U_i - \frac{K}{n} \sum\limits_{i=1}^{n} X_i \tag{5}$$

In the Formula (5), $i$ is the sampling sequence, $i = 1, 2, 3, \ldots, n$, and $n$ is the number of points of the input angular rate.

The WMMCRG prototype was fixed on the turntable test system for scale factor testing. The rate sensitivity was measured under a rotating disk at input angular rates of 0 (°)/s, ±0.1 (°)/s, ±0.2 (°)/s, ±0.5 (°)/s, ±1 (°)/s, ±2 (°)/s, ±5 (°)/s, ±10 (°)/s, ±20 (°)/s, ±50 (°)/s, ±100 (°)/s, ±150 (°)/s, and ±200 (°)/s, and the output values at each point were recorded. The results of this test are shown in Figure 10.

$$U = 0.02001\ \Omega + 0.034$$

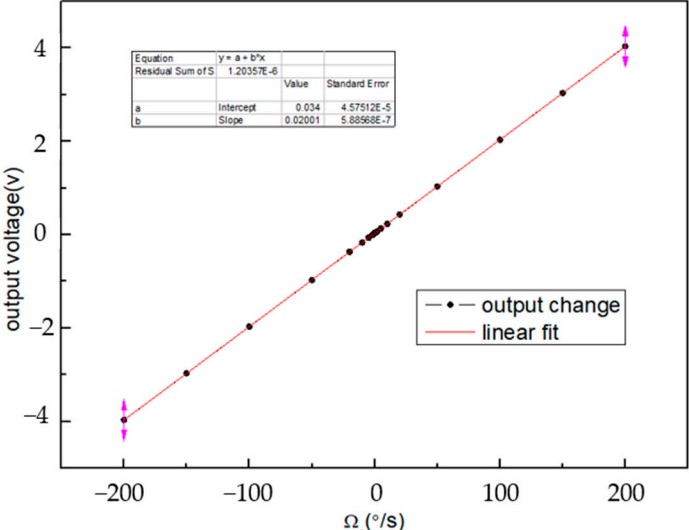

**Figure 10.** WMMCRG scale factor test result.

### 5.3. Static Test

The output of the gyroscope is recorded, the input axis of the gyroscope is parallel to the geography from east to west, the sampling rate and sampling time of the gyroscope test are set, and the output $U_{ot}$ of the gyroscope in the sampling time is recorded after the preparation time.

In the experiment, the sampling rate is 1 Hz, the sampling time is 3000 s, and the output of the gyroscope is shown in Figure 11.

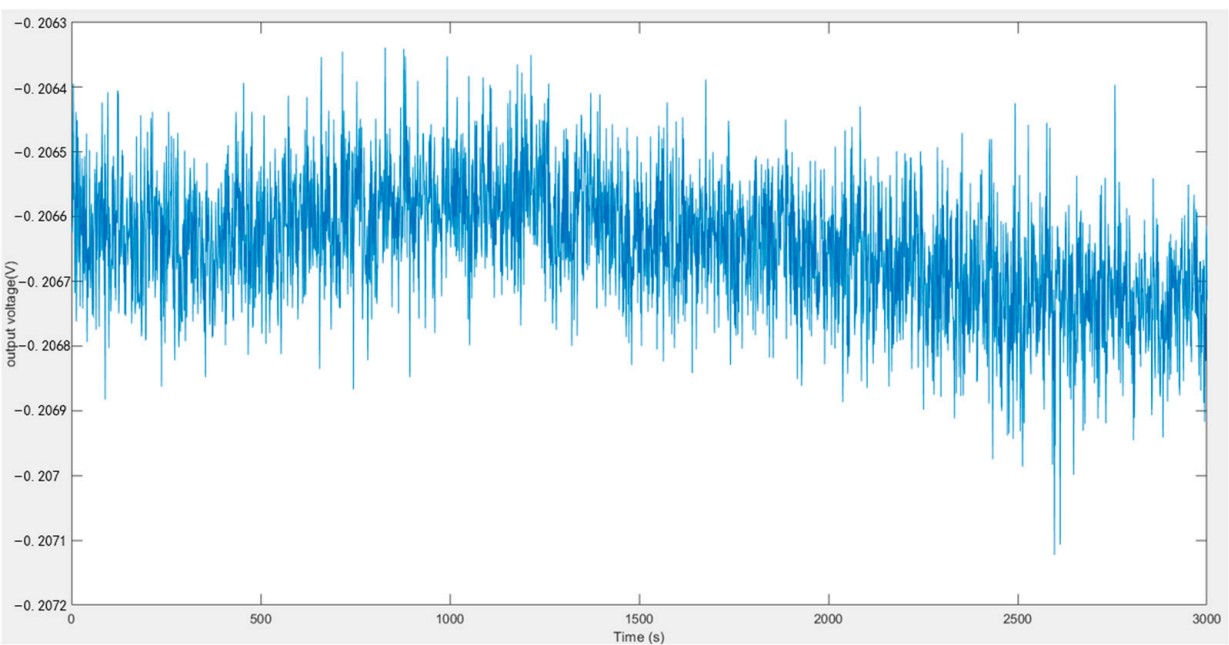

**Figure 11.** WMMCRG static output curve.

The zero bias Formula (6) of the gyroscope is as follows:

$$B_0 = \frac{1}{K} \cdot \frac{1}{n} \sum_{i=1}^{n} U_{ot} \tag{6}$$

In the Formula (6), $B_0$ is the zero bias of the gyroscope, $K$ is the scale factor, $n$ is the sampling time, and $U_{ot}$ is the $i$-th sampling value.

The zero bias of the gyroscope is calculated to be $10.3323°/\text{s}$.

The measured output values of the gyroscope are averaged according to the prescribed sampling time (smoothing time), and the zero bias stability of the gyroscope ($1\sigma$ method) is obtained by calculating the standard deviation of these averages.

Calculate the number of samples $M$ included in the specified smoothing time according to the Formula (7):

$$M = T \cdot f_t \tag{7}$$

where $M$ is the number of samples contained in the specified smoothing time, $T$ is the specified smoothing time, and $f_t$ is the sampling rate.

The initial sample is divided into a group per $M$, and the m group ($m > 6$) is obtained, and the $B_{mi}$ is calculated according to the Formula (8):

$$B_{mi} = \frac{1}{K} \cdot \frac{1}{M} \sum_{j=1}^{M} U_{o[(i-1)M+j]} \tag{8}$$

In the Formula (8), $B_{mi}$ is the zero bias sample ($i$ is an integer), $K$ is the scale factor, $U_{oi}$ is the output of the $i$-th test gyroscope, and $j$ is the serial number of samples in each group.

The zero bias stability $B_{s\sigma}$ of gyroscope is calculated according to the Formula (9):

$$B_{s\sigma} = \sqrt{\frac{1}{m-1} \sum_{i=1}^{m} (B_{mi} - B_0)^2} \tag{9}$$

In the Formula (9), $B_{s\sigma}$ is the gyroscope zero bias stability ($1\sigma$ method), $B_0$ is the gyroscope zero bias, and $m$ is the number of sample groups.

The zero bias stability of the gyroscope is calculated to be $10.869°/\text{h}$.

The Allan variance analysis results of the WMMCRG output are shown in Figure 12. From the Allan variance curve, it can be determined that the bias instability is approximately 1.974 (°)/h and the angle random walk is approximately 16 (°)/√h.

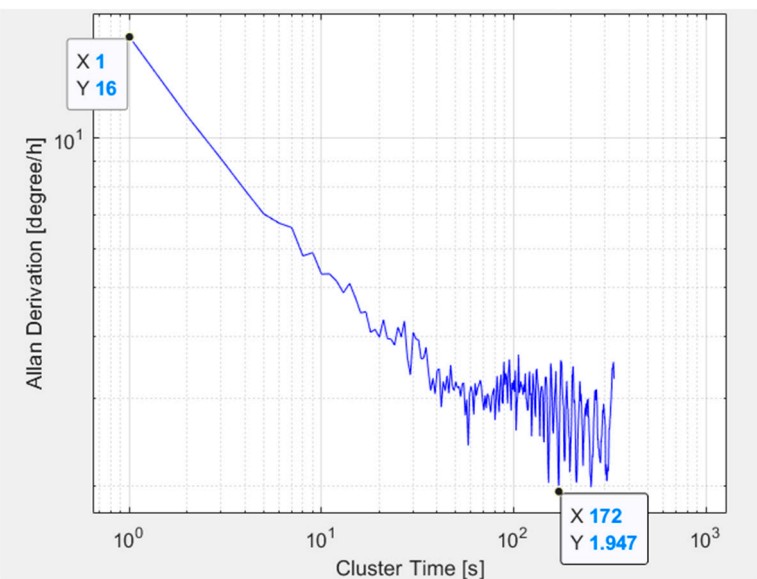

**Figure 12.** WMMCRG Allan derivation result.

## 6. Conclusions and Discussion

In this paper, a new type of gyroscope structure is proposed, and its processing mode and control method are described. The structure has the advantages of small frequency split, high output sensitivity, simple structure, and small control error. The structure of the sensor is simulated, and the simulation results show that the frequency split of the sensor in the working mode is very small, and the coupling effect of other modes on the working mode is also very small. Through the analog circuit, the driving closed loop, the sensing closed loop, and the quadrature error correction closed loop are established on two PCBs. The experimental results prove the superiority of the new structure and closed loop system. The bias instability is 1.974 (°)/h, the bias stability is 10.869°/h, the zero bias is 10.3323°/s, the angle random walk is 16 (°)/√h, and the frequency splitting is 0.02 Hz. The scale factor of the gyroscope and the static parameters of the gyroscope are improved. The experimental results prove the correctness of the structural design method proposed in this paper.

**Author Contributions:** Conceptualization, X.G. and H.C.; methodology, X.G.; software, R.C.; validation, W.W., Q.C. and S.Y.; formal analysis, W.W.; investigation, X.G.; resources, X.G.; data curation, C.S.; writing—original draft preparation, X.G.; writing—review and editing, X.G.; visualization, R.C.; supervision, C.S.; project administration, H.C.; funding acquisition, H.C. All authors have read and agreed to the published version of the manuscript.

**Funding:** This work is supported by National Key Research and Development Program of China (No.2022YFB3205000), National Natural Science Foundation of China (No. U2230206), Technology Field Fund of Basic Strengthening Plan of China (2020JCJQJJ409, 2021-JCJQ-JJ-0315), National defense basic scientific research program (WDZC20190303) and Pre-Research Field Foundation of Equipment Development Department of China (No. 80917010501). The research is also supported by Fundamental Research Program of Shan-xi Province (20210302123020 and 20210302123062), Shanxi province key laboratory of quantum sensing and precision measurement (201905D121001), Key Research and Development (R&D) Projects of Shanxi Province (202003D111004), the Aeronautical Science Foundation of China (2019080U0002), Beijing Key Laboratory of High Dynamic Navigation Technology open fund (HDN2021102) and the Fund for Shanxi "1331 Project" Key Subjects Construction.

**Institutional Review Board Statement:** Not applicable.

**Informed Consent Statement:** Not applicable.

**Data Availability Statement:** Not applicable.

**Conflicts of Interest:** The authors declare no conflict of interest.

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
