# Peer review of "Design and Experiment for N = 3 Wineglass Mode Metal Cylindrical Resonator Gyroscope Closed-Loop System"

_electronics, doi:10.3390/electronics12010131_

Round 1

Reviewer 1 Report

Overall, the paper is OK but I feel the authors could significantly improve the quality and the presentation of the research. Many of the figures are nearly unreadable as the font size is way to small on figure axis's (i.e. Figures 15-17). In addition, Figures 6 and 7 really don't add any value to this paper. Figure 12 and 13 are also not real useful. It would be nice to know what the application for this device is as systems that I am familiar with, have allan derivations many orders of magnitude better than what is presented in this paper, thus applications may explain the reasoning behind this effort. The fabrication is easy but the device is very large compared to many structures that I work with. 

There are a lot of other metrics that could and should be presented in this device such as bias stability, reliability, addition comparison between the two devices. You stated the bias instability but it would be good to show a plot of these measurements verse just stating the numbers.

I am also not convinced as to why you try to compare the two devices using the same fabrication approach since it is not optimal for the CVG device. Any structure where the fabrication is tailored to fit one device over another is bound to appear to be the best. I do realize you are trying to present the ease of fabrication for these but how useful will a device like this be? 

Lastly, a little English word usage and cleanup would also be helpful. 

Author Response

Thank you!

Reviewer 2 Report

The authors have demonstrated their finding of gyro structure based on Mode Metal Cylindrical Resonator Gyroscope. I have following observations in this manuscript.

1.      In general abbreviations are kept inside brackets after abbreviating properly. However I found that  “Wineglass Mode Metal Cylindrical Resonator Gyroscope (TWMG)” as an error. Therefore authors need to correct the same in the abstract.

2.      Introduction section need to improved massively as many latest finding on the topic is still missing.

3.      I found that the mathematical expressions used here has already been used widely by researchers, however, I found no citation of the source. Do correct it.

4.      The captions of the figures should start with capital letters only.

5.      Figures 15, 16 and 17 is not visible clearly. The texts are very small it needs to be improved.

6.      The results are not providing a clear explanation. The experimental setups and steps also needed to explain more.

In the light of above observations, I find this article for consideration in this journal upon changes as suggested.

Author Response

Thank you!

Reviewer 3 Report

The paper is clearly presented and the results are consistent.

Instead of just claiming that the performances of their gyroscope are superior, I would suggest the authors to add quantitative comparisons with results of other researchers, by citing appropriate figures and references.

Please, specify better the meaning of a and b in the differential system (1.1). What does it mean "Among them"? Actually, the terms after (1.1) should be intended as definitions!

Please, explain in which way (1.1) has been computationally solved in order  to determine the modes?

Author Response

Thank you!
